# Learning Sparse Latent Representations with the Deep Copula Information Bottleneck

**Aleksander Wieczorek**,\* **Mario Wieser**,\* **Damian Murezzan, Volker Roth**
University of Basel, Switzerland
`{firstname.lastname}@unibas.ch`

## Abstract

Deep latent variable models are powerful tools for representation learning. In this paper, we adopt the deep information bottleneck model, identify its shortcomings and propose a model that circumvents them. To this end, we apply a copula transformation which, by restoring the invariance properties of the information bottleneck method, leads to disentanglement of the features in the latent space. Building on that, we show how this transformation translates to sparsity of the latent space in the new model. We evaluate our method on artificial and real data.

## 1 Introduction

In recent years, deep latent variable models (Kingma & Welling, 2013; Rezende et al., 2014; Goodfellow et al., 2014) have become a popular toolbox in the machine learning community for a wide range of applications (Ledig et al., 2016; Reed et al., 2016; Isola et al., 2016). At the same time, the compact representation, sparsity and interpretability of the latent feature space have been identified as crucial elements of such models. In this context, multiple contributions have been made in the field of relevant feature extraction (Chalk et al., 2016; Alemi et al., 2016) and learning of disentangled representations of the latent space (Chen et al., 2016; Bouchacourt et al., 2017; Higgins et al., 2017).

In this paper, we consider latent space representation learning. We focus on disentangling features with the copula transformation and, building on that, on forcing a compact low-dimensional representation with a sparsity-inducing model formulation. To this end, we adopt the *deep information bottleneck (DIB)* model (Alemi et al., 2016) which combines the information bottleneck and variational autoencoder methods. The *information bottleneck (IB)* principle (Tishby et al., 2000) identifies relevant features with respect to a target variable. It takes two random vectors $x$ and $y$ and searches for a third random vector $t$ which, while compressing $x$, preserves information contained in $y$. A *variational autoencoder (VAE)* (Kingma & Welling, 2013; Rezende et al., 2014) is a generative model which learns a latent representation $t$ of $x$ by using the variational approach.

Although DIB produces good results in terms of image classification and adversarial attacks, it suffers from two major shortcomings. First, the IB solution only depends on the copula of $x$ and $y$ and is thus invariant to strictly monotone transformations of the marginal distributions. DIB does not preserve this invariance, which means that it is unnecessarily complex by also implicitly modelling the marginal distributions. We elaborate on the fundamental issues arising from this lack of invariance in Section 3. Second, the latent space of the IB is not sparse which results in the fact that a compact feature representation is not feasible.

Our contribution is two-fold: In the first step, we restore the invariance properties of the information bottleneck solution in the DIB. We achieve this by applying a transformation of $x$ and $y$ which makes the latent space only depend on the copula. This is a way to fully represent all the desirable features inherent to the IB formulation. The model is also simplified by ensuring robust and fully non-parametric treatment of the marginal distributions. In addition, the problems arising from the lack of invariance to monotone transformations of the marginals are solved. In the second step, once the invariance properties are restored, we exploit the sparse structure of the latent space of DIB. This is possible thanks to the copula transformation in conjunction with using the sparse parametrisation

---

\*These authors contributed equally.

of the information bottleneck, proposed by (Rey et al., 2014). It translates to a more compact latent space that results in a better interpretability of the model.

The remainder of this paper is structured as follows: In Section 2, we review publications on related models. Subsequently, in Section 3, we describe the proposed copula transformation and show how it fixes the shortcomings of DIB, as well as elaborate on the sparsity induced in the latent space. In Section 4, we present results of both synthetic and real data experiments. We conclude our paper in Section 5.

## 2 RELATED WORK

The IB principle was introduced by (Tishby et al., 2000). The idea is to compress the random vector $x$ while retaining the information of the random vector $y$. This is achieved by solving the following variational problem: $\min_{p(t|x)} I(x;t) - \lambda I(t;y)$, with the assumption that $y$ is conditionally independent of $t$ given $x$, and where $I$ stands for mutual information. In recent years, copula models were combined with the IB principle in (Rey & Roth, 2012) and extended to the sparse meta-Gaussian IB (Rey et al., 2014) to become invariant against strictly monotone transformations. Moreover, the IB method has been applied to the analysis of deep neural networks in (Tishby & Zaslavsky, 2015), by quantifying mutual information between the network layers and deriving an information theoretic limit on DNN efficiency.

The variational bound and reparametrisation trick for autoencoders were introduced in (Kingma & Welling, 2013; Rezende et al., 2014). The variational autoencoder aims to learn the posterior distribution of the latent space $p(t|x)$ and the *decoder* $p(x|t)$. The general idea of combining the two approaches is to identify the solution $t$ of the information bottleneck with the latent space $t$ of the variational autoencoder. Consequently, the terms $I(x;t)$ and $I(t;y)$ in the IB problem can be expressed in terms of the parametrised conditionals $p(t|x)$, $p(y|t)$.

Variational lower bounds on the information bottleneck optimisation problem have been considered in (Chalk et al., 2016) and (Alemi et al., 2016). Both approaches, however, treat the differential entropy of the marginal distribution as a positive constant, which is not always justified (see Section 3). A related model is introduced in (Pereyra et al., 2017), where a penalty on the entropy of output distributions of neural networks is imposed. These approaches do not introduce the invariance against strictly monotone transformations and thus do not address the issues we identify in Section 3.

A sizeable amount of work on modelling the latent space of deep neural networks has been done. The authors of (Alvarez & Salzmann, 2016) propose the use of a group sparsity regulariser. Other techniques, e.g. in (Mozer & Smolensky, 1989) are based on removing neurons which have a limited impact on the output layer, but they frequently do not scale well with the overall network size. More recent approaches include training neural networks of smaller size to mimic a deep network (Hinton et al., 2015; Romero et al., 2014). In addition, multiple contributions have been proposed in the area of latent space disentanglement (Chen et al., 2016; Bouchacourt et al., 2017; Higgins et al., 2017; Denton & Birodkar, 2017). None of the approaches consider the influence of the copula on the modelled latent space.

Copula models have been proposed in the context of Bayesian variational methods in (Suh & Choi, 2016), (Tran et al., 2015) and (Han et al., 2016). The former approaches focus on treating the latent space variables as indicators of local approximations of the original space. None of the three approaches relate to the information bottleneck framework.

## 3 MODEL

### 3.1 FORMULATION

In order to specify our model, we start with a parametric formulation of the information bottleneck:

$$\max_{\phi,\theta} -I_\phi(t;x) + \lambda I_{\phi,\theta}(t;y), \tag{1}$$

where $I$ stands for mutual information with its parameters in the subscript. A parametric form of the conditionals $p_\phi(t|x)$ and $p_\theta(y|t)$ as well as the information bottleneck Markov chain $t - x - y$ are assumed. A graphical illustration of the proposed model is depicted in Figure 1.

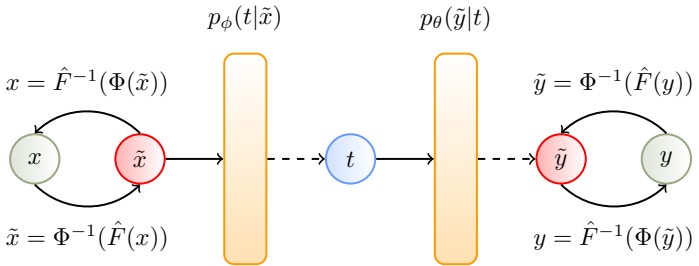

Figure 1: Deep information bottleneck with the copula augmentation. Green circles describe random variables and orange rectangles denote deep nets parametrising the random variables. The blue circle stands for latent random variables whereas the red circle denotes the copula transformed random variables.

The two terms in Eq. (1) have the following forms:

$$I_\phi(T; X) = D_{KL}\left(p(t|x)p(x)\|p(t)p(x)\right) = \mathbb{E}_{p(x)}D_{KL}\left(p_\phi(t|x)\|p(t)\right), \tag{2}$$

and

$$I_{\phi,\theta}(T; Y) = D_{KL}\left(\left[\int p(t|y, x)p(y, x)\,\mathrm{d}x\right]\|p(t)p(y)\right)$$
$$= \mathbb{E}_{p(x,y)}\mathbb{E}_{p_\phi(t|x)}\log p_\theta(y|t) + h(Y), \tag{3}$$

because of the Markov assumption in the information bottleneck model $p_\phi(t|x, y) = p_\phi(t|x)$. We denote with $h(y) = -\mathbb{E}_{p(y)}[\log p(y)]$ the *entropy* for discrete $y$ and the *differential entropy* for continuous $y$. We then assume a conditional independence copula and Gaussian margins:

$$p_\phi(t|x) = c_{T|X}(u(t|x)|x) \prod_j p_{\phi_j}(t_j|x) = \prod_j N(t_j|\mu_j(x), \sigma_j^2(x))$$

where $t_j$ is the $j$th marginal of $t = (t_1, \ldots, t_d)$, $c_{t|x}$ is the copula density of $t|x$, $u(t|x) := F_{t|x}(t|x)$ is the uniform density indexed by $t|x$, and the functions $\mu_j(x), \sigma_j^2(x)$ are implemented by deep networks. We make the same assumption about $p_\theta(y|t)$.

## 3.2 MOTIVATION

As we stated in Section 1, the deep information bottleneck model derived in Section 3.1 is not invariant to strictly increasing transformations of the marginal distributions. The IB method is formulated in terms of mutual information $I(x, y)$, which depends only on the copula and therefore does not depend on monotone transformations of the marginals: $I(x, y) = MI(x, y) - MI(x) - MI(y)$, where $MI(x)$, for $x = (x_1, \ldots, x_d)$, denotes the multi-information, which is equal to the negative copula entropy, as shown by Ma & Sun (2011):

$$MI(X) := D_{KL}(p(x)\|\prod_j p_j(x_j)) = \int c_X(u(x)) \log c_X(u(x))\mathrm{d}u = -h(c_X(u(x))). \tag{4}$$

**Issues with lack of invariance to marginal transformations.**

1. On the encoder side (Eq. (2)), the optimisation is performed over the parametric conditional margins $p_\phi(t_j|x)$ in $I_\phi(t; x) = \mathbb{E}_{p(x)}D_{KL}\left(p_\phi(t|x)\|p(t)\right)$. When a monotone transformation $x_j \to \tilde{x}_j$ is applied, the required invariance property can only be guaranteed if the model for $\phi$ (in our case a deep network) is flexible enough to compensate for this transformation, which can be a severe problem in practice (see example in Section 4.1).

2. On the decoder side, assuming Gaussian margins in $p_\theta(y_j|t)$ might be inappropriate for modelling $y$ if the domain of $y$ is not equal to the real numbers, e.g. when $y$ is defined only on a bounded

interval. If used in a generative way, the model might produce samples outside the domain of $y$. Even if other distributions than Gaussian are considered, such as truncated Gaussian, one still needs to make assumptions concerning the marginals. According to the IB formulation, such assumptions are unnecessary.

3. Also on the decoder side, we have: $I_\phi(t; y) = \mathbb{E}_{p(x,y)}\mathbb{E}_{p_\phi(t|x)} \log p_\theta(y|t) + h(y)$. The authors of Alemi et al. (2016) argue that since $h(y)$ is constant, it can be ignored in computing $I_\phi(t; y)$. This is true for a fixed or for a discrete $y$, but not for the class of monotone transformations of $y$, which should be the case for a model specified with mutual informations only. Since the left hand side of this equation ($I_\phi(t; y)$) is invariant against monotone transformations, and $h(y)$ in general depends on monotone transformations, the first term on the right hand side ($\mathbb{E}_{p(x,y)}\mathbb{E}_{p_\phi(t|x)} \log p_\theta(y|t)$) cannot be invariant to monotone transformations. In fact, under such transformations, the differential entropy $h(y)$ can take any value from $-\infty$ to $+\infty$, which can be seen easily by decomposing the entropy into the copula entropy and the sum of marginal entropies (here, $j$ stands for the $j$th dimension):

$$h(y) = h(c_y(u(y))) + \sum_j h(y_j) = -MI(y) + \sum_j h(y_j). \tag{5}$$

The first term (i.e. the copula entropy which is equal to the negative multi-information, as in Eq. (4)) is a non-positive number. The marginal entropies $h(y_j)$ can take any value when using strictly increasing transformations (for instance, the marginal entropy of a uniform distribution on $[a, b]$ is $\log(b - a)$). As a consequence, the entropy term $h(y)$ in Eq. (3) can be treated as a constant only either for one specific $y$ or for discrete $y$, but not for all elements of the equivalence class containing all monotone transformations of $y$. Moreover, every such transformation would lead to different $(I(x, t), I(y, t))$ pairs in the information curve, which basically makes this curve arbitrary. Thus, $h(y)$ being constant is a property that needs to be restored.

## 3.3 PROPOSED SOLUTION

The issues described in Section 3.2 can be fixed by using transformed variables (for a $d$ dimensional $x = (x_1, \ldots, x_d)$, $x_j$ stands for the $j$th dimension):

$$\tilde{x}_j = \Phi^{-1}(\hat{F}(x_j)), \quad x_j = \hat{F}^{-1}(\Phi(\tilde{x}_j)), \tag{6}$$

where $\Phi$ is the Gaussian cdf and $\hat{F}$ is the empirical cdf. The same transformation is applied to $y$. In the copula literature, these transformed variables are sometimes called *normal scores*. Note that the mapping is (approximately) invertible: $x_j = \hat{F}^{-1}(\Phi(\tilde{x}_j))$, with $\hat{F}^{-1}$ being the empirical quantiles treated as a function (e.g. by linear interpolation). This transformation fixes the invariance problem on the encoding side (issue 1), as well as the problems on the decoding side: problem 2 disappeared because the transformed variables $\tilde{x}_j$ are standard normal distributed, and problem 3 disappeared because the decoder part (Eq. (3)) now has the form:

$$\mathbb{E}_{p(\tilde{x},\tilde{y})}\mathbb{E}_{p_\phi(t|\tilde{x})} \log p_\theta(\tilde{y}|t) = I_\phi(t; \tilde{y}) + MI(\tilde{y}) - \sum_j h(\tilde{y}_j) = I_\phi(t; \tilde{y}) - h(c_{\text{inv}}(u(\tilde{y}))) \tag{7}$$

where $c_{\text{inv}}(u(\tilde{y}))$ is indeed constant for all strictly increasing transformations applied to $y$.

Having solved the IB problem in the transformed space, we can go back to the original space by using the inverse transformation according to Eq. (6) $x_j = \hat{F}^{-1}(\Phi(\tilde{x}_j))$. The resulting model is thus a variational autoencoder with $x$ replaced by $\tilde{x}$ in the first term and $y$ replaced by $\tilde{y}$ in the second term.

**Technical details.** We assume a simple prior $p(t) = \mathcal{N}(t; 0, I)$. Therefore, the KL divergence $D_{KL}(p_\phi(t|\tilde{x})\|p(t))$ is a divergence between two Gaussian distributions and admits an analytical form. We then estimate

$$I(t; \tilde{x}) = \mathbb{E}_{p(\tilde{x})}D_{KL}(p_\phi(t|\tilde{x})\|p(t)) \approx \frac{1}{n} \sum_i D_{KL}(p_\phi(t|\tilde{x}_i)\|p(t)) \tag{8}$$

and all the gradients on (mini-)batches.

For the decoder side, $\mathbb{E}_{p(\tilde{x},\tilde{y})}\mathbb{E}_{p_\phi(t|\tilde{x})} \log p_\theta(\tilde{y}|t)$ is needed. We train our model using the backpropagation algorithm. However, this algorithm can only handle deterministic nodes. In order to overcome

this problem, we make use of the reparametrisation trick (Kingma & Welling, 2013; Rezende et al., 2014):

$$I(t; \tilde{y}) = \mathbb{E}_{p(\tilde{x}, \tilde{y})} \mathbb{E}_{\epsilon \sim \mathcal{N}(0, I)} \sum_j \log p_\theta(\tilde{y}_j | t = \vec{\mu}_j(\tilde{x}) + diag(\sigma_j(\tilde{x}))\epsilon) + \text{const.,} \qquad (9)$$

with $\tilde{y}_j = \Phi^{-1}(\hat{F}(y_j))$.

### 3.4 SPARSITY OF THE LATENT SPACE

In this section we explain how the sparsity constraint on the information bottleneck along with the copula transformation result in sparsity of the latent space $t$. We first introduce the Sparse Gaussian Information Bottleneck and subsequently show how augmenting it with the copula transformation leads to the sparse $t$.

**Sparse Gaussian Information Bottleneck.**   Recall that the information bottleneck compresses $x$ to a new variable $t$ by minimising $I(x; t) - \lambda I(t; y)$. This ensures that some amount of information with respect to a second "relevance" variable $y$ is preserved in the compression.

The assumption that $x$ and $y$ are jointly Gaussian-distributed leads to the *Gaussian Information Bottleneck* (Chechik et al., 2005) where the solution $t$ can be proved to also be Gaussian distributed. In particular, if we denote the marginal distribution of $x$: $x \sim \mathcal{N}(0, \Sigma_x)$, the optimal $t$ is a noisy projection of $x$ of the following form:

$$t = Ax + \xi, \quad \xi \sim \mathcal{N}(0, I) \quad \Rightarrow \quad t|x \sim \mathcal{N}(Ax, I), \quad t \sim \mathcal{N}(0, A\Sigma_x A^\top + I).$$

The mutual information between $x$ and $t$ is then equal to: $I(x; t) = \frac{1}{2} \log |A\Sigma_x A^\top + I|$.

In the *sparse* Gaussian Information Bottleneck, we additionally assume that $A$ is diagonal, so that the compressed $t$ is a sparse version of $x$. Intuitively, sparsity follows from the observation that for a pair of random variables $x, x'$, any full-rank projection $Ax'$ of $x'$ would lead to the same mutual information since $I(x, x') = I(x; Ax')$, and a reduction in mutual information can only be achieved by a rank-deficient matrix $A$. For diagonal projections, this immediately implies sparsity of $A$.

**Sparse latent space of the Deep Information Bottleneck.**   We now proceed to explain the sparsity induced in the latent space of the *copula version* of the DIB introduced in Section 3.3. We will assume a possibly general, abstract pre-transformation of $x$, $f_\beta$, which accounts for the encoder network along with the copula transformation of $x$. Then we will show how allowing for this abstract pre-transformation, in connection with the imposed sparsity constraint of the sparse information bottleneck described above, translates to the sparsity of the latent space of the copula DIB. By sparsity we understand the number of active neurons in the last layer of the encoder.

To this end, we use the Sparse Gaussian Information Bottleneck model described above. We analyse the encoder part of the DIB, described with $I(x, t)$. Consider the general Gaussian Information Bottleneck (with $x$ and $y$ jointly Gaussian and a full matrix $A$) and the deterministic pre-transformation, $f_\beta(x)$, performed on $x$. The pre-transformation is parametrised by a set of parameters $\beta$, which might be weights of neurons should $f_\beta$ be implemented as a neural network. Denote by $M$ a $n \times p$ matrix which contains $n$ i.i.d. samples of $Af_\beta(x)$, i.e. $M = AZ$ with $Z = (f_\beta(x_1), \ldots, f_\beta(x_n))^\top$. The optimisation of mutual information $I(x, t)$ in $\min I(x; t) - \lambda I(t; y)$ is then performed over $M$ and $\beta$.

Given $f_\beta$ and the above notation, the estimator of $I(x; t) = \frac{1}{2} \log |A\Sigma_x A^\top + I|$ becomes:

$$\hat{I}(x; t) = \frac{1}{2} \log \left| \frac{1}{n} MM^\top + I \right|, \qquad (10)$$

which would further simplify to $\hat{I}(x; t) = \frac{1}{2} \sum_i \log(D_{ii} + 1)$, if the pre-transformation $f_\beta$ were indeed such that $D := \frac{1}{n} MM^\top$ were diagonal. This is equivalent to the Sparse Gaussian Information Bottleneck model described above. Note that this means that the sparsity constraint in the Sparse Gaussian IB does not cause any loss of generality of the IB solution as long as the abstract

pre-transformation $f_\beta$ makes it possible to diagonalise $\frac{1}{n}MM^\top$ in Eq. (10). We can, however, approximate this case by forcing this diagonalisation in Eq. (10), i.e. by only considering the diagonal part of the matrix: $I'(x;t) = \frac{1}{2}\log\left|\text{diag}(\frac{1}{n}MM^\top + I)\right|$.

We now explain why this approximation (replacing $\hat{I}(x;t)$ with $I'(x;t)$) is justified and how it leads to $f_\beta$ finding a low-dimensional representation of the latent space. Note that for any positive definite matrix $B$, the determinant $|B|$ is always upper bounded by $\prod_i B_{ii} = |\text{diag}(B)|$, which is a consequence of Hadamard's inequality. Thus, instead of minimising $\hat{I}(x;t)$, we minimise an upper bound $I'(x;t) \geq \hat{I}(x;t)$ in the Information Bottleneck cost function. Equality is obtained if the transformation $f_\beta$, which we assume to be part of an "end-to-end" optimisation procedure, indeed successfully diagonalised $D = \frac{1}{n}MM^\top$. Note that equality in the Hadamard's inequality is equivalent to $D + I$ being orthogonal, thus $f_\beta$ is forced to find the "most orthogonal" representation of the inputs in the latent space. Using a highly flexible $f_\beta$ (for instance, modelled by a deep neural network), we might approximate this situation reasonably well. This explains how the copula transformation translates to a low-dimensional representation of the latent space.

We indeed see disentanglement and sparse structure of the latent space learned by the copula DIB model by comparing it to the plain DIB without the copula transformation. We demonstrate it in Section 4.

## 4 EXPERIMENTS

We now proceed to experimentally verify the contributions of the copula Deep Information Bottleneck. The goal of the experiments is to test the impact of the copula transformation. To this end, we perform a series of pair-wise experiments, where DIB without and with (cDIB) the copula transformation are tested in the same set-up. We use two datasets (artificial and real-world) and devise multiple experimental set-ups.

### 4.1 ARTIFICIAL DATA

First, we construct an artificial dataset such that a high-dimensional latent space is needed for its reconstruction (the dataset is reconstructed when samples from the latent space spatially coincide with it in its high-dimensional space). We perform monotone transformations on this dataset and test the difference between DIB and cDIB on reconstruction capabilities as well as classification predictive score.

**Dataset and set-up.** The model used to generate the data consists of two input vectors $x_1$ and $x_2$ drawn form a uniform distribution defined on $[0, 2]$ and vectors $k_1$ and $k_2$ drawn uniformly from $[0, 1]$. Additional inputs are $x_{i=3...10} = a_i * k_1 + (1 - a_i) * k_2 + 0.3 * b_i$ with $a_i, b_i$ drawn from a uniform distribution defined on $[0, 1]$. All input vectors $x_{1...10}$ form the input matrix $X$. Latent variables $z_1 = \sqrt{x_1^2 + x_2^2}$ and $z_2 = z_1 + x_4$ are defined and then normalised by dividing through their maximum value. Finally, random noise is added. Two target variables $y_1 = z_2 * \cos(1.75 * \pi * z_1)$ and $y_2 = z_2 * \sin(1.75 * \pi * z_1)$ are then calculated. $y_1$ and $y_2$ form a spiral if plotted in two dimensions. The angle and the radius of the spiral are highly correlated, which leads to the fact that a one-dimensional latent space can only reconstruct the backbone of the spiral. In order to reconstruct the details of the radial function, one has to use a latent space of at least two dimensions. We generate 200k samples from $X$ and $y$. $X$ is further transformed to beta densities using strictly increasing transformations. We split the samples into test (20k samples) and training (180k samples) sets. The generated samples are then transformed with the copula transformation (Eq. (6)) to $\tilde{X}$ and $\tilde{y}$ and split in the same way into test and training sets. This gives us the four input sets $X_{train}, X_{test}, \tilde{X}_{train}, \tilde{X}_{test}$ and the four target sets $y_{train}, y_{test}, \tilde{y}_{train}, \tilde{y}_{test}$.

We use a latent layer with ten nodes that model the means of the ten-dimensional latent space $t$. The variance of the latent space is set to 1 for simplicity. The encoder as well as the decoder consist of a neural network with two fully-connected hidden layers with 50 nodes each. We use the softplus function as the activation function. Our model is trained using mini batches (size = 500) with the Adam optimiser (Kingma & Ba, 2014) for 70000 iterations using a learning rate of 0.0006.

**Experiment 1.** In the first experiment, we compare the information curves produced by the DIB and its copula augmentation (Figure 2(a)). To this end, we use the sets $(X_{train}, y_{train})$ and $(\tilde{X}_{train}, \tilde{y}_{train})$ and record the values of $I(x;t)$ and $I(y;t)$ while multiplying the $\lambda$ parameter every 500 iterations by 1.06 during training. One can observe an increase in the mutual information from approximately 6 in the DIB to approximately 11 in the copula DIB. At the same time, only two dimensions are used in the latent space $t$ by the copula DIB. The version without copula does not provide competitive results despite using 10 out of 18 dimensions of the latent space $t$. In Appendix B, we extend this experiment to comparison of information curves for other pre-processing techniques as well as to subjecting the training data to monotonic transformations other than the beta transformation.

**Experiment 2.** Building on Experiment 1, we use the trained models for assessing their predictive quality on test data $(X_{test}, y_{test})$ and $(\tilde{X}_{test}, \tilde{y}_{test})$. We compute predictive scores of the latent space $t$ with respect to the generated $y$ in the form of mutual information $I(t;y)$ for all values of the parameter $\lambda$. The resulting information curve shows an increased predictive capability of cDIB in Figure 2(b) and exhibits no difference to the information curve produced in Experiment 1. Thus, the increased mutual information reported in Experiment 1 cannot only be attributed to overfitting.

**Experiment 3.** In the third experiment, we qualitatively assess the reconstruction capability of cDIB compared to plain DIB (Figure 3). We choose the value of $\lambda$ such that in both models two dimensions are active in the latent space. Figure 3(b) shows a detailed reconstruction of $y$. The reconstruction quality of plain DIB on test data results in a tight backbone which is not capable of reconstructing $y$ (Figure 3(a)).

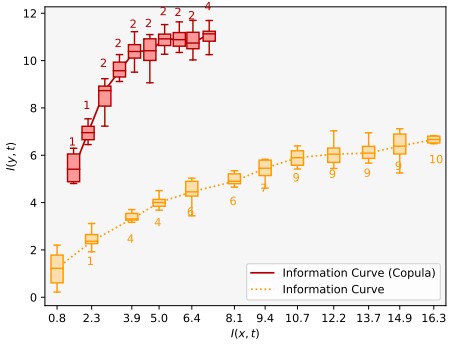
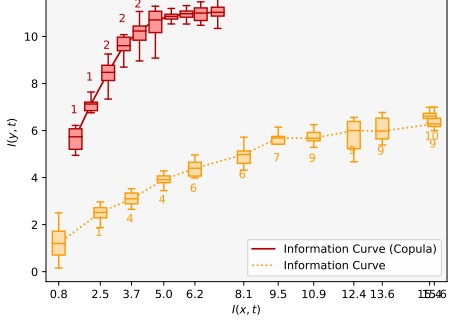

(a) Training Information Curve

(b) Predictive Information Curve (evaluated on test data)

Figure 2: Information curves for the artificial experiment. The red curve describes the information curve with copula transformation whereas the orange one illustrates the plain information curve. The numbers represent the dimensions in the latent space $t$ which are needed to reconstruct the output $y$.

**Experiment 4.** We further inspect the information curves of DIB and cDIB by testing how the copula transformation adds resilience of the model against outliers and adversarial attacks in the training phase. To simulate an adversarial attack, we randomly choose 5% of all entries in the datasets $X_{train}$ and $\tilde{X}_{train}$ and replace them with outliers by adding uniformly sampled noise within the range [1,5]. We again compute information curves for the training procedure and compare normal training with training with data subject to an attack for the copula and non-copula models. The results (Figure 4(a)) show that the copula model is more robust against outlier data than the plain one. We attribute this behaviour directly to the copula transformation, as ranks are less sensitive to outliers than raw data.

**Experiment 5.** In this experiment, we investigate how the copula transformation affects convergence of the neural networks making up the DIB. We focus on the encoder and track the values

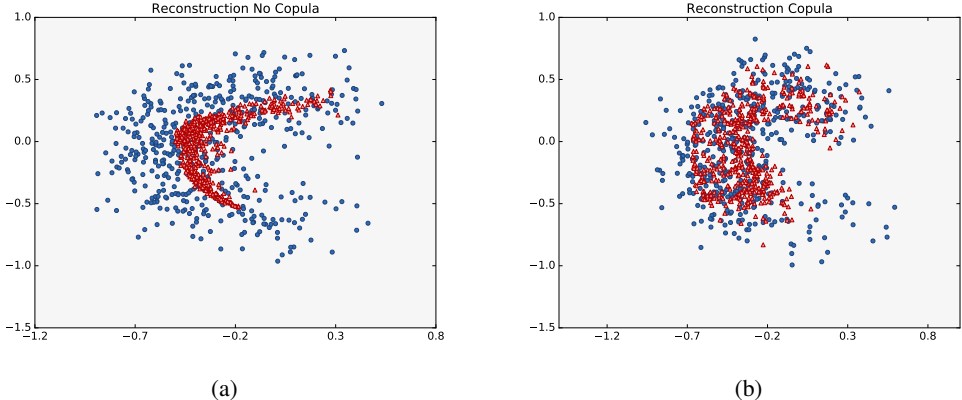

Figure 3: Reconstruction of $y$ without (a) and with the copula transformation (b). Blue circles depict the output space and the red triangles — the conditionals means $\mu(y)$. The better the red triangles fill up the blue area, the better the reconstruction.

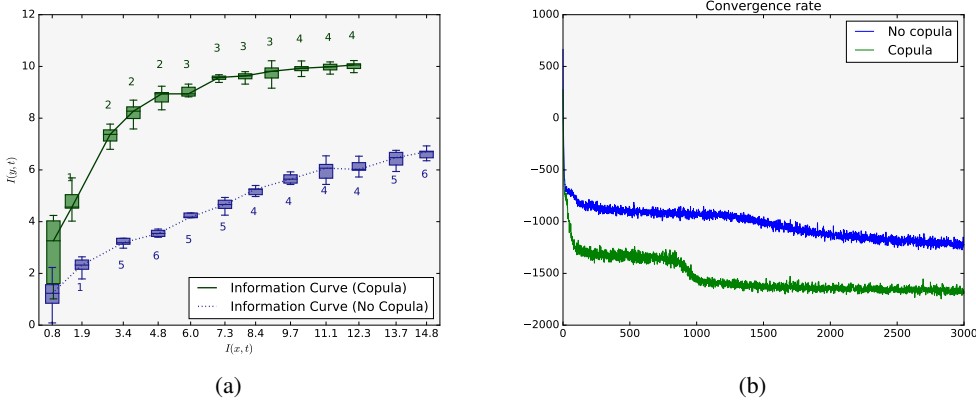

Figure 4: Information curves for training with outlier data (a) and a sample convergence plot of DIB and cDIB models for $\lambda = 100$ (b). The $y$ axis of the right figure shows the value of the loss function.

of the loss function. Figure 4(b) shows a sample comparison of convergence of DIB and cDIB for $\lambda = 100$. One can see that the cDIB starts to converge around iteration no. 1000, whereas the plain DIB takes longer. This can be explained by the fact that in the copula model the marginals are normalised to the same range of normal quantiles by the copula transformation. This translates to higher convergence rates.

## 4.2 REAL-WORLD DATA

We continue analysing the impact of the copula transformation on the latent space of the DIB with a real-world dataset. We first report information curves analogous to Experiment 1 (Section 4.1) and proceed to inspect the latent spaces of both models along with sensitivity analysis with respect to $\lambda$.

**Dataset and Set-up.** We consider the unnormalised *Communities and Crime* dataset Lyons et al. (1998) from the UCI repository[1]. The dataset consisted of 125 predictive, 4 non-predictive and 18 target variables with 2215 samples in total. In a preprocessing step, we removed all missing values from the dataset. In the end, we used 1901 observations with 102 predictive and 18 target variables in our analysis.

---

[1]http://archive.ics.uci.edu/ml/datasets/communities+and+crime+unnormalized

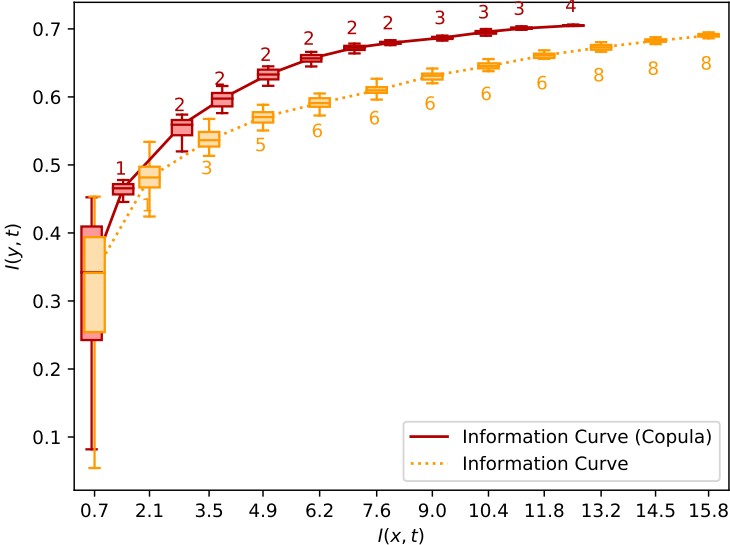

Figure 5: Information curves for the real data experiment. The red curve is the information curve with copula transformation whereas the orange one depicts the plain information curve. The numbers represent the dimensions in the latent space $t$ which are needed to reconstruct the output $y$.

We use a latent layer with 18 nodes that models the means of the 18-dimensional latent space $t$. Again, the variance of the latent and the output space is set to 1. The stochastic encoder as well as the stochastic decoder consist of a neural network with two fully-connected hidden layers with 100 nodes each. Softplus is employed as the activation function. The decoder uses a Gaussian likelihood. Our model is trained for 150000 iterations using mini batches with a size of 1255. As before, we use Adam (Kingma & Ba, 2014) with a learning rate of 0.0005.

**Experiment 6.** Analogously to Experiment 1 (Section 4.1), information curves stemming from the DIB and cDIB models have been computed. We record the values of $I(x; t)$ and $I(y; t)$ while multiplying the $\lambda$ parameter every 500 iterations by 1.01 during training. Again, the information curve for the copula model yields larger values of mutual information, which we attribute to the increased flexibility of the model, as we pointed out in Section 3.3. In addition, the application of the copula transformation leads to a much lower number of used dimensions in the latent space. For example, copula DIB uses only four dimensions in the latent space for the highest $\lambda$ values. DIB, on the other hand, needs eight dimensions in the latent space and nonetheless results in lower mutual information scores. In order to show that our information curves are significantly different, we perform a Kruskal-Wallis rank test (p-value of $1.6 * 10^{-16}$).

**Experiment 7.** This experiment illustrates the difference in the disentanglement of the latent spaces of the DIB model with and without the copula transformation. We select two variables which yielded highest correlation with the target variable *arsons* and plot them along with their densities. In order to obtain the corresponding class labels (rainbow colours in Figure 6), we separate the values of *arsons* in eight equally-sized bins. A sample comparison of latent spaces of DIB and cDIB for $\lambda = 21.55$ is depicted in Figure 6. A more in-depth analysis of sensitivity of the learned latent space to $\lambda$ is presented in Appendix A. The latent space $t$ of DIB appears consistently less structured than that of cDIB, which is also reflected in the densities of the two plotted variables. In contrast, we can identify a much clearer structure in the latent space with respect to our previously calculated class labels.

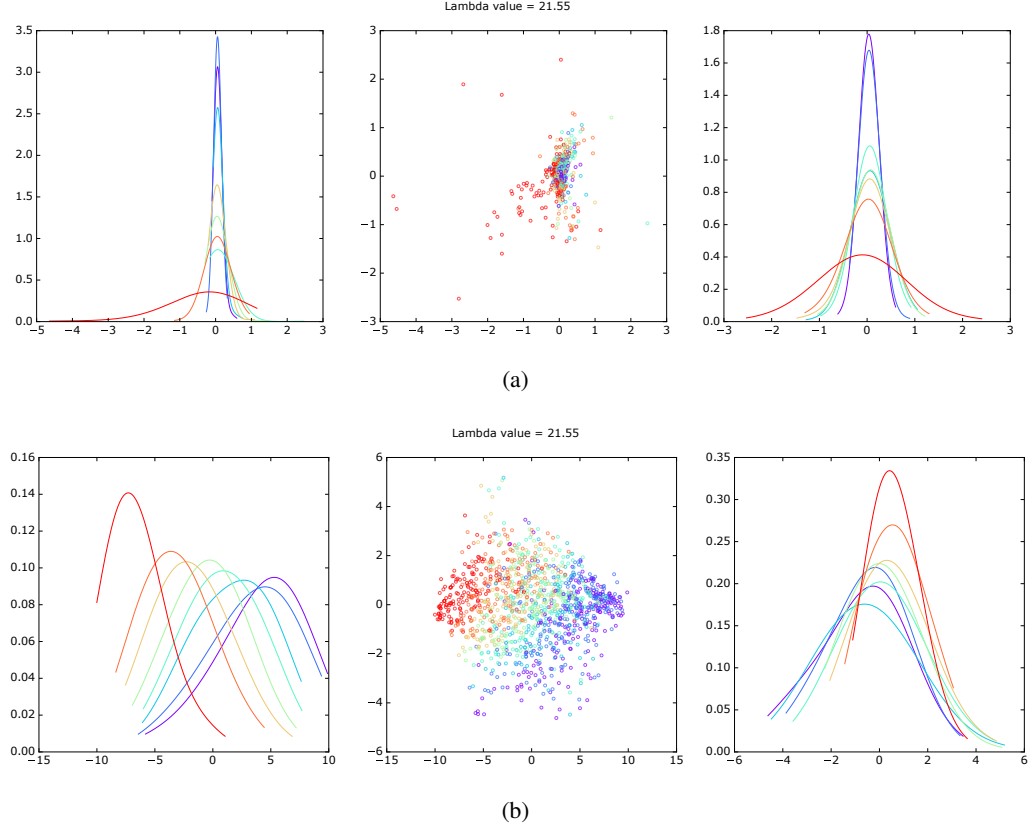

Figure 6: Latent space $t$ consisting of two dimensions along with marginal densities without (a) and with (b) the copula transformation. The copula transformation leads to a disentangled latent space, which is reflected in non-overlapping modes of marginal distributions.

## 5 CONCLUSION

We have presented a novel approach to compact representation learning of deep latent variable models. To this end, we showed that restoring invariance properties of the Deep Information Bottleneck with a copula transformation leads to disentanglement of the features in the latent space. Subsequently, we analysed how the copula transformation translates to sparsity in the latent space of the considered model. The proposed model allows for a simplified and fully non-parametric treatment of marginal distributions which has the advantage that it can be applied to distributions with arbitrary marginals. We evaluated our method on both artificial and real data. We showed that in practice the copula transformation leads to latent spaces that are disentangled, have an increased prediction capability and are resilient to adversarial attacks. All these properties are not sensitive to the only hyperparameter of the model, $\lambda$.

In Section 3.2, we motivated the copula transformation for the Deep Information Bottleneck with the lack of invariance properties present in the original Information Bottleneck model, making the copula augmentation particularly suited for the DIB. The relevance of the copula transformation, however, reaches beyond the variational autoencoder, as evidenced by e.g. resilience to adversarial attacks or the positive influence on convergence rates presented in Section 4. These advantages of our model that do not simply follow from restoring the Information Bottleneck properties to the DIB, but are additional benefits of the copula. The copula transformation thus promises to be a simple but powerful addition to the general deep learning toolbox.

ACKNOWLEDGMENTS

This work was partially supported by the Swiss National Science Foundation under grants CR32I2_159682 and 51MRP0_158328 (SystemsX.ch).

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

# A    SENSITIVITY OF THE LATENT SPACE TO $\lambda$

We augment Experiment 7 from Section 4 with sensitivity analysis of the latent space with respect to the chosen value of the only hyperparameter, $\lambda$. To this end, we recompute Experiment 7 for different values of $\lambda$ ranging between $1.79$ and $1897.15$ (which corresponds to the reported information curves). The results are reported in Figures 7 and 8. As can be seen, the latent space of the copula DIB is consistently better structured then that of the plain DIB.

# B    EXTENSION OF EXPERIMENT 1

Building on Experiment 1 from Section 4, we again compare the information curves produced by the DIB and its copula augmentation. We compare the copula transformation with data normalisation (transformation to mean 0 and variance 1) in Figure 9(a). We also replace the beta transformation with gamma in the experimental set-up described in Section 4 and report the results in Figure 9(b). As in Experiment 1, one con see that the information curve for the copula version of DIB lies above the plain one. The latent space uses fewer dimensions as well.

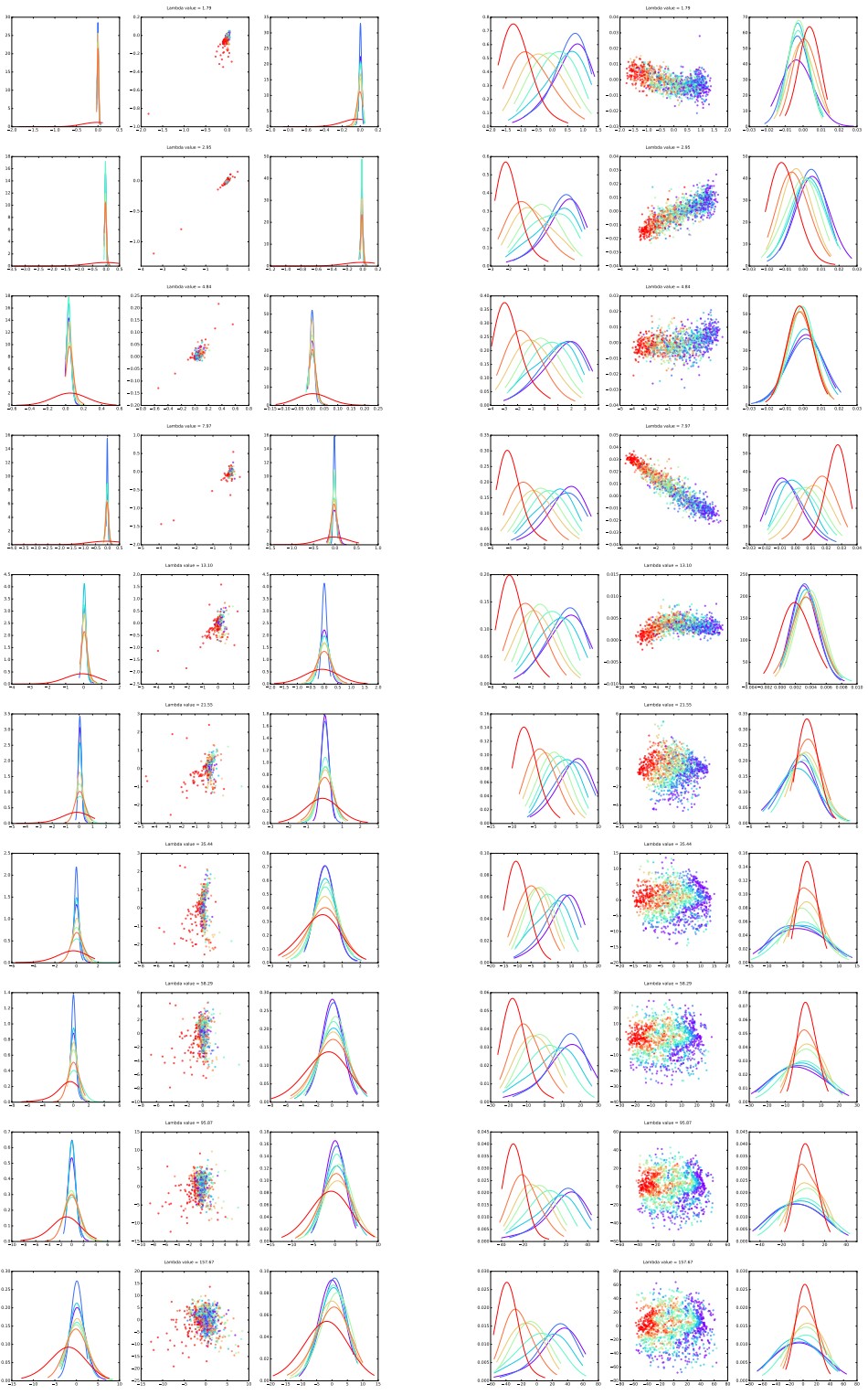

Figure 7: Latent space $t$ consisting of two dimensions along with marginal densities without (left) and with (right) the copula transformation, for different values of $\lambda$ ranging between 1.79 and 95.87. The copula transformation leads to a disentangled latent space, which is reflected in non-overlapping modes of marginal distributions.

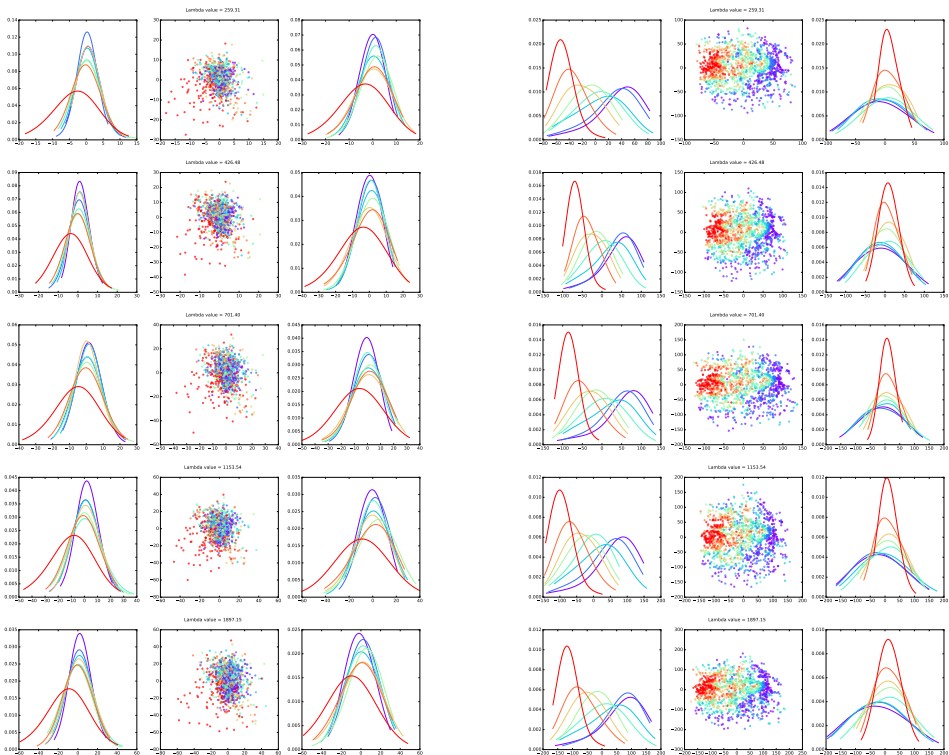

Figure 8: Continuation of Figure 7. Latent space $t$ consisting of two dimensions along with marginal densities without (left) and with (right) the copula transformation, for different values of $\lambda$ ranging between $157.67$ and $1897.15$. The copula transformation leads to a disentangled latent space, which is reflected in non-overlapping modes of marginal distributions.

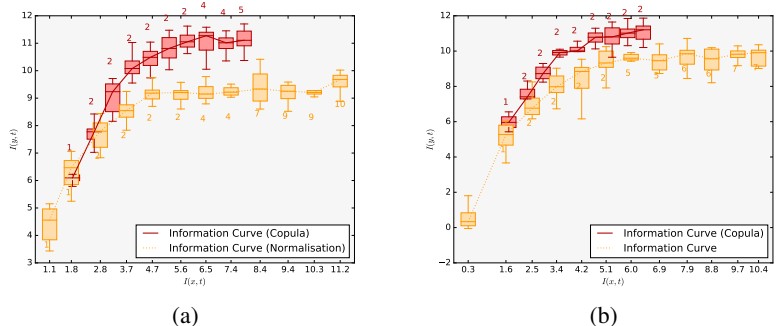

Figure 9: Different extensions of Experiment 1: (a) comparison of the copula transformation to normalising the input data (to zero mean and unit variance), (b) Experiment 1 with a *gamma* instead of a *beta* transformation. All curves are computed over the same range of the hyperparameter $\lambda$. The copula pre-transformation yields higher information curves and uses fewer dimensions in the latent space.

