# OpenReview forum: "Learning Sparse Latent Representations with the Deep Copula Information Bottleneck"
_ICLR.cc/2018/Conference — Accept (Poster)_

### Official Review · AnonReviewer2 · 2017-11-26
**Interesting work . Both the clarity and the experimental results have been improved in the revised version**

**Rating:** 6
**Confidence:** 3

**Review:**

This paper presents a sparse latent representation learning algorithm based on an information theoretic objective formulated through meta-Gaussian information bottleneck and solved via variational auto-encoder stochastic optimization. The authors suggest Gaussianify the data using copula transformation and  further adopt a diagonal determinant approximation with justification of minimizing an upper bound of mutual information.  Experiments include both artificial data and real data.

The paper is unclear at some places and writing gets confusing. For example, it is unclear whether and when explicit or implicit transforms are used for x and y in the experiments, and the discussion at the end of Section 3.3 also sounds confusing. It would be more helpful if the author can make those points more clear and offer some guidance about the choices between explicit and implicit transform in practice. Moreover, what is the form of f_beta and how beta is optimized?  In the first equation on page 5, is tilde y involved? How to choose lambda?

If MI is invariant to monotone transformations and information curves are determined by MIs, why “transformations basically makes information curve arbitrary”? Can you elaborate?

Although the experimental results demonstrate that the proposed approach with copula transformation yields higher information curves, more compact representation and better reconstruction quality, it would be more significant if the author can show whether these would necessarily lead to any improvements on other goals such as classification accuracy or robustness under adversarial attacks.

Minor comments:

- What is the meaning of the dashed lines and the solid lines respectively in Figure 1?
- Section 3.3 at the bottom of page 4: what is tilde t_j? and x in the second term? Is there a typo?
- typo, find the “most orthogonal” representation if the inputs -> of the inputs

Overall, the main idea of this paper is interesting and well motivated and but the technical contribution seems incremental. The paper suffers from lack of clarity at several places and the experimental results are convincing but not strong enough.

***************
Updates:
***************
The authors have clarified some questions that I had and further demonstrated the benefits of copula transform with new experiments in the revised paper. The new results are quite informative and addressed some of the concerns raised by me and other reviewers. I have updated my score to 6 accordingly.

---

### Official Review · AnonReviewer1 · 2017-11-26
**An extension to DVIB**

**Rating:** 6
**Confidence:** 3

**Review:**

[====================================REVISION ======================================================]
Ok so the paper underwent major remodel, which significantly improved the clarity. I do agree now on Figure 5, which tips the scale for me to a weak accept.
[====================================END OF REVISION ================================================]

This paper explores the problems of existing Deep variational bottle neck approaches for compact representation learning. Namely, the authors adjust deep variational bottle neck to conform to invariance properties (by making latent variable space to depend on copula only) - they name this model a  copula extension to dvib. They then go on to explore the sparsity of the latent space

My main issues with this paper are experiments: The proposed approach is tested only on 2 datasets (one synthetic, one real but tiny - 2K instances) and some of the plots (like Figure 5) are not convincing to me. On top of that, it is not clear how two methods compare computationally and how introduction of the copula  affects the convergence (if it does)

Minor comments
Page 1: forcing an compact -> forcing a compact
“and and” =>and
Section 2: mention that I is mutual information, it is not obvious for everyone

Figure 3: circles/triangles are too small, hard to see
Figure 5: not really convincing. B does not appear much more structured than a, to me it looks like a simple transformation of a.

---

### Official Review · AnonReviewer3 · 2017-11-28
**This paper improved on an existing latent variable model by combining ideas from different but somewhat related papers. Experimental results indeed show some improvements.**

**Rating:** 6
**Confidence:** 1

**Review:**

The paper proposed a copula-based modification to an existing deep variational information bottleneck model, such that the marginals of the variables of interest (x, y) are decoupled from the DVIB latent variable model, allowing the latent space to be more compact when compared to the non-modified version. The experiments verified the relative compactness of the latent space, and also qualitatively shows that the learned latent features are more 'disentangled'. However, I wonder how sensitive are the learned latent features to the hyper-parameters and optimizations?

Quality: Ok. The claims appear to be sufficiently verified in the experiments. However, it would have been great to have an experiment that actually makes use of the learned features to make predictions. I struggle a little to see the relevance of the proposed method without a good motivating example.

Clarity: Below average. Section 3 is a little hard to understand. Is q(t|x) in Fig 1 a typo? How about t_j in equation (5)? There is a reference that appeared twice in the bibliography (1st and 2nd).

Originality and Significance: Average. The paper (if I understood it correctly) appears to be mainly about borrowing the key ideas from Rey et. al. 2014 and applying it to the existing DVIB model.

---

### Official Review · AnonReviewer5 · 2018-01-15
**A promising improvement to DVIB, but paper suffers from lack of clarity and limited experimentation.**

**Rating:** 5
**Confidence:** 4

**Review:**

This paper identifies and proposes a fix for a shortcoming of the Deep Information Bottleneck approach, namely that the induced representation is not invariant to monotonic transform of the marginal distributions (as opposed to the mutual information on which it is based). The authors address this shortcoming by applying the DIB to a transformation of the data, obtained by a copula transform. This explicit approach is shown on synthetic experiments to preserve more information about the target, yield better reconstruction and converge faster than the baseline. The authors further develop a sparse extension to this Deep Copula Information Bottleneck (DCIB), which yields improved representations (in terms of disentangling and sparsity) on a UCI dataset.

(significance) This is a promising idea. This paper builds on the information theoretic perspective of representation learning, and makes progress towards characterizing what makes for a good representation. Invariance to transforms of the marginal distributions is clearly a useful property, and the proposed method seems effective in this regard.
Unfortunately, I do not believe the paper is ready for publication as it stands, as it suffers from lack of clarity and the experimentation is limited in scope.

(clarity) While Section 3.3 clearly defines the explicit form of the algorithm (where data and labels are essentially pre-processed via a copula transform), details regarding the “implicit form” are very scarce. From Section 3.4, it seems as though the authors are optimizing the form of the gaussian information bottleneck I(x,t), in the hopes of recovering an encoder $f_\beta(x)$ which gaussianizes the input (thus emulating the explicit transform) ? Could the authors clarify whether this interpretation is correct, or alternatively provide additional clarifying details ? There are also many missing details in the experimental section: how were the number of “active” components selected ? Which versions of the algorithm (explicit/implicit) were used for which experiments ? I believe explicit was used for Section 4.1, and implicit for 4.2 but again this needs to be spelled out more clearly. I would also like to see a discussion (and perhaps experimental comparison) to standard preprocessing techniques, such as PCA-whitening.

(quality) The experiments are interesting and seem well executed. Unfortunately, I do not think their scope (single synthetic, plus a single UCI dataset) is sufficient. While the gap in performance is significant on the synthetic task, this gap appears to shrink significantly when moving to the UCI dataset. How does this method perform for more realistic data, even e.g. MNIST ? I think it is crucial to highlight that the deficiencies of DIB matter in practice, and are not simply a theoretical consideration. Similarly, the representation analyzed in Figure 7 is promising, but again the authors could have targeted other common datasets for disentangling, e.g. the simple sprites dataset used in the beta-VAE paper. I would have also liked to see a more direct and systemic validation of the claims made in the paper. For example, the shortcomings of DIB identified in Section 3.1, 3.2 could have been verified more directly by plotting I(y,t) for various monotonic transformations of x. A direct comparison of the explicit and implicit forms of the algorithms would also also make for a stronger paper in my opinion.

Pros:
* Theoretically well motivated
* Promising results on synthetic task
* Potential for impact
Cons:
* Paper suffers from lack of clarity (method and experimental section)
* Lack of ablative / introspective experiments
* Weak empirical results (small or toy datasets only).

---

> ### Author Response · Authors · 2018-01-19
> **Additional review response part 1**
>
> We would like to thank the reviewer for the additional review. We respond to the questions and issues raised in the review below.
>
>
>
>
> While Section 3.3 clearly defines the explicit form of the algorithm (where data and labels are essentially pre-processed via a copula transform), details regarding the “implicit form” are very scarce. From Section 3.4, it seems as though the authors are optimizing the form of the gaussian information bottleneck I(x,t), in the hopes of recovering an encoder $f_\beta(x)$ which gaussianizes the input (thus emulating the explicit transform) ? Could the authors clarify whether this interpretation is correct, or alternatively provide additional clarifying details?
>
> This seems to be a misunderstanding. The $f_\beta$ transformation stands for an abstract, general transformation of the input data. In our model, it is implemented by the copula transformation (explicit or implicit) and the encoder network. $f_\beta$ thus does not emulate the explicit transformation, and is not confined to representing the (implicit or explicit) copula transformation. The copula transformation, not necessarily implemented as a neural network, is a part of $f_\beta$.
> The purpose of introducing $f_\beta$ is to explain the difference of the model with and without the extra copula transformation and why applying the transformation translates to sparsity not observed in the “regular” sparse Gaussian information bottleneck.
> We elaborate on the difference between the implicit and explicit copula in the answer to the last question.
>
>
>
>
> There are also many missing details in the experimental section: how were the number of “active” components selected ?
>
> The only parameter of our model is $\lambda$. As described in Section 3.4, by continuously increasing $\lambda$, one decreases sparsity defined by the number of active neurons. Thus, one can adjust the number of active components by continuously varying $\lambda$ (curves in Figures 2, 4, 6 with increasing numbers of active components correspond to increasing $\lambda$).
> The number of active components is chosen differently in different experiments. In Experiments 1, 6, 7 $\lambda$, and thus the number of active components, is varied over a large interval. In Experiment 3, $\lambda$ is also varied, and subsequently chosen so that the dimensionality of latent spaces in the two compared models is the same.
>
>
>
>
> Which versions of the algorithm (explicit/implicit) were used for which experiments ? I believe explicit was used for Section 4.1, and implicit for 4.2 but again this needs to be spelled out more clearly
>
> As we mentioned in the rebuttal, throughout the paper as well as for the experiments, the explicit copula transformation defined in Eq. (6) is used. The explicit transformation is also the default choice of the form of the copula transformation.
>
>
>
>
> I would also like to see a discussion (and perhaps experimental comparison) to standard preprocessing techniques, such as PCA-whitening.
>
> PCA whitening, in contrast to the copula transformation, does not disentangle marginal distributions from the dependence structure captured by the copula. It also does not restore the invariance properties of the model we identified as motivation. It does not lead to a boost in information curves such as in Figure 2; we can add the appropriate experiment to our manuscript.
>
>
>
>
> I do not think their [experiments’] scope (single synthetic, plus a single UCI dataset) is sufficient. While the gap in performance is significant on the synthetic task, this gap appears to shrink significantly when moving to the UCI dataset. How does this method perform for more realistic data, even e.g. MNIST ? I think it is crucial to highlight that the deficiencies of DIB matter in practice, and are not simply a theoretical consideration.
> […]
> the representation analyzed in Figure 7 is promising, but again the authors could have targeted other common datasets for disentangling, e.g. the simple sprites dataset used in the beta-VAE paper.
>
> We would like to stress that imposing sparsity on the latent representation is an important aspect of our model. It is in general difficult to quantify latent representations. Our model yields significantly sparser representations even when the information curves are closer.
> Our model shows its full strength when a multiview analysis is involved, especially with data where multiple variables have different and rescaled distributions. Datasets constructed such that marginals (or simply labels, such as in the MNIST dataset) are uniform distributed do not pose enough challenge, since the output space is too easy to reconstruct even without the copula transformation.
> As for dataset size, we would like to point out that finding meaningful sparse representations is more challenging for smaller datasets with higher dimensionality, therefore we think that the datasets we used do show the most relevant properties of the copula DIB.

---

> ### Author Response · Authors · 2018-01-19
> **Additional review response part 2**
>
> I would have also liked to see a more direct and systemic validation of the claims made in the paper. For example, the shortcomings of DIB identified in Section 3.1, 3.2 could have been verified more directly by plotting I(y,t) for various monotonic transformations of x.
>
> We verified this for beta transformation in Experiment 1. We observe that the impact of our method is most pronounced when different variables are transformed in possibly different ways (i.e. when they are subject to diverse transformations with various scales).
>
>
>
>
> A direct comparison of the explicit and implicit forms of the algorithms would also also make for a stronger paper in my opinion.
>
> We mention the implicit copula transformation learned by neural networks in Section 3.3 for completeness as an alternative to the default explicit approach, but we would like to point out that the explicit approach is a preferred choice in practice.
> In the same section (in the revised paper), we elaborate on the few situations where the implicit copula might be advantageous, such as when there is a necessity of implicit tie breaking between data points. We also explain why the explicit copula is usually more advantageous. One circumvents the problem of devising an architecture capable of learning the marginal cdf, thus simplifying the neural network. Perhaps more importantly, the implicit approach does not scale well with dimensionality of the data, since the networks used for approximating the marginal cdf have to be trained independently for every dimension.

---

### Decision · Program_Chairs · 2018-01-29
**ICLR 2018 Conference Acceptance Decision**

**Decision:**

Accept (Poster)

**Comment:**

Observing that in contrast to classical information bottleneck, the deep variational information bottleneck (DVIB) model is not invariant to monotonic transformations of input and output marginals, the authors show how to incorporate this invariance along with sparsity in DVIB using the copula transform. The revised version of the paper addressed some of the reviewer concerns about clarity as well as the strength of the experimental section, but the authors are encouraged to improve these aspects of the paper further.